# Further development and feasibility randomised controlled trial of a digital programme for adolescent depression, MoodHwb: study protocol

Rhys Bevan Jones [1,2,3] Sally Merry [4] Paul Stallard [5]
Elizabeth Randell [6] Bryony Weavers [1,3] Anna Gray [1,3] Elaine Hindle,[7]
Marcela Gavigan,[7] Samantha Clarkstone,[6] Rhys Williams-Thomas [6]
Vince Poile [6] Rebecca Playle [6] Jonathan I Bisson [1,3]
Rachel McNamara [6] Frances Rice [1,3] Sharon Anne Simpson [7]

FR and SAS are joint senior authors.

For numbered affiliations see end of article.

**Correspondence to**
Dr Rhys Bevan Jones;
bevanjonesr1@cardiff.ac.uk

## ABSTRACT

**Introduction** A digital programme, MoodHwb, was codesigned with young people experiencing or at high risk of depression, parents/carers and professionals, to provide support for young people with their mood and well-being. A preliminary evaluation study provided support for the programme theory and found that MoodHwb was acceptable to use. This study aims to refine the programme based on user feedback, and to assess the acceptability and feasibility of the updated version and study methods.

**Methods and analysis** Initially, this study will refine MoodHwb with the involvement of young people, including in a pretrial acceptability phase. This will be followed by a multicentre feasibility randomised controlled trial comparing MoodHwb plus usual care with a digital information pack plus usual care. Up to 120 young people aged 13–19 years with symptoms of depression and their parents/carers will be recruited through schools, mental health services, youth services, charities and voluntary self-referral in Wales and Scotland. The primary outcomes are the feasibility and acceptability of the MoodHwb programme (including usage, design and content) and of trial methods (including recruitment and retention rates), assessed 2 months postrandomisation. Secondary outcomes include potential impact on domains including depression knowledge and stigma, help-seeking, well-being and depression and anxiety symptoms measured at 2 months postrandomisation.

**Ethics and dissemination** The pretrial acceptability phase was approved by the Cardiff University School of Medicine Research Ethics Committee (REC) and the University of Glasgow College of Medicine, Veterinary and Life Sciences REC. The trial was approved by Wales NHS REC 3 (21/WA/0205), the Health Research Authority(HRA), Health and Care Research Wales (HCRW), university health board Research and Development (R&D) departments in Wales, and schools in Wales and Scotland. Findings will be disseminated in peer-reviewed open-access journals, at conferences and meetings, and online to academic, clinical, and educational audiences and the wider public.

**Trial registration number** ISRCTN12437531.

## STRENGTHS AND LIMITATIONS OF THIS STUDY

⇒ MoodHwb was codesigned with young people with lived experience or at high risk of depression, parents/carers and practitioners.
⇒ MoodHwb is bilingual, personalised according to user needs and preferences, and has been developed in line with evidence-based approaches to support young people with depressive symptoms and their parents/carers, friends and practitioners.
⇒ Multiple methods will be used to collect data in a range of settings for the trial, including validated questionnaires, semistructured interviews, a focus group, and Web/app usage monitoring.
⇒ As this is a feasibility study, we will not be able to determine the effectiveness of MoodHwb; however, the findings will inform the design of a definitive effectiveness trial.
⇒ Participants will not be blind to treatment allocation as this is not possible with an intervention of this kind.

## BACKGROUND

Depression is common in adolescence and is associated with social and educational impairments, deliberate self-harm and suicide. It can also mark the beginning of long-term mental health difficulties. Subthreshold depressive symptoms can also affect quality of life and are a risk indicator for depressive disorder. Early treatment and prevention of adolescent depression is, therefore, a major public health concern.[1 2] There are also concerns about the economic burden of mental disorders.[3 4] However, most adolescents with depression do not access or receive formal help and engaging young people (YP) in prevention and early intervention programmes is challenging.[5–7]

Guidelines for the prevention and management of depression in YP (eg, National Institute for Health and Care Excellence (NICE)[8]; American Academy of Child & Adolescent Psychiatry[9]) stress the need for good information and evidence-based psychosocial interventions for YP and families/carers, and digital interventions are recommended for mild depression in UK NICE guidelines.[8] Digital mental health technologies (ie, resources and interventions) are a potential way of increasing access to support, at relatively low cost. Most YP in high-income countries have access to the internet and mobile devices, including in the UK.[10 11] Many in low-income and middle-income countries can also access the internet, where there are few alternative sources of support.[12] While there is increasing evidence for the effectiveness of technologies for adolescent depression,[13 14] more programmes are needed to support adolescents that are accessible, evidence based, developed with user input and evaluated in line with research frameworks—and to provide greater options for YP in accessing help.[15]

MoodHwb is a bilingual (English and Welsh), interactive and multiplatform digital programme designed to support YP with their mood and well-being—and their families/carers, friends and practitioners (figure 1).

It was codesigned using a series of interviews and focus groups with YP experiencing or at high risk of depressive symptoms (due to family history), parents/carers and professionals, alongside workshops with a digital media company and an animator.[16–18] The development was also informed by a systematic review,[19] design, educational and psychological theories, and a preliminary 'logic model' (programme theory) (figure 2).

As part of the developmental work, there was an early evaluation of MoodHwb with a small number of YP and their parents/carers and practitioners, recruited via primary and secondary Child and Adolescent Mental Health Services (CAMHS), school counsellors and nurses, and a previous study sample (Early Prediction of Adolescent Depression (EPAD) study).[20] Preliminary results were encouraging, and participants found the

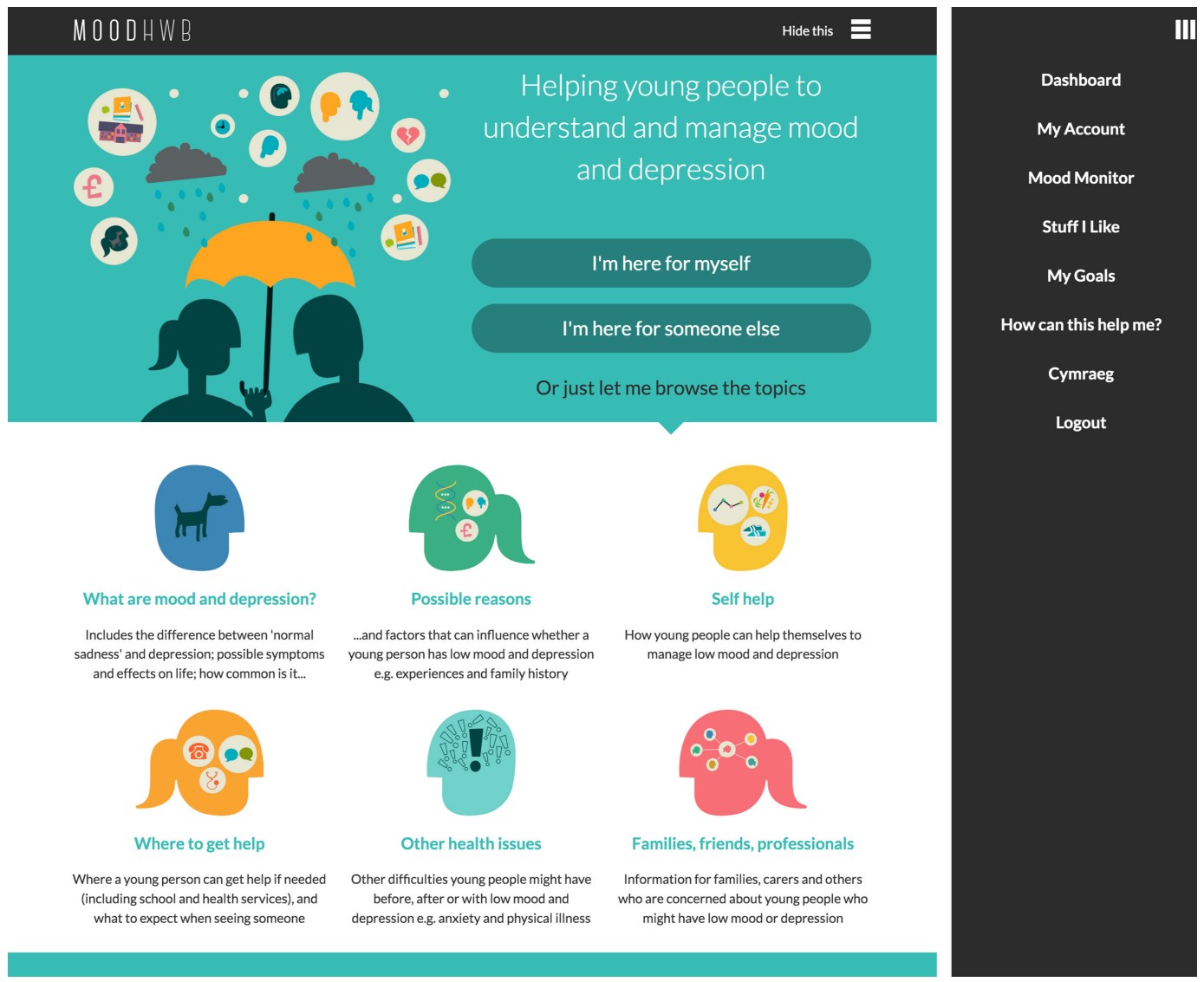

**Figure 1** MoodHwb (v1) welcome screen (main image/left) and open menu (right) (from Bevan Jones *et al*[20] 2020a).

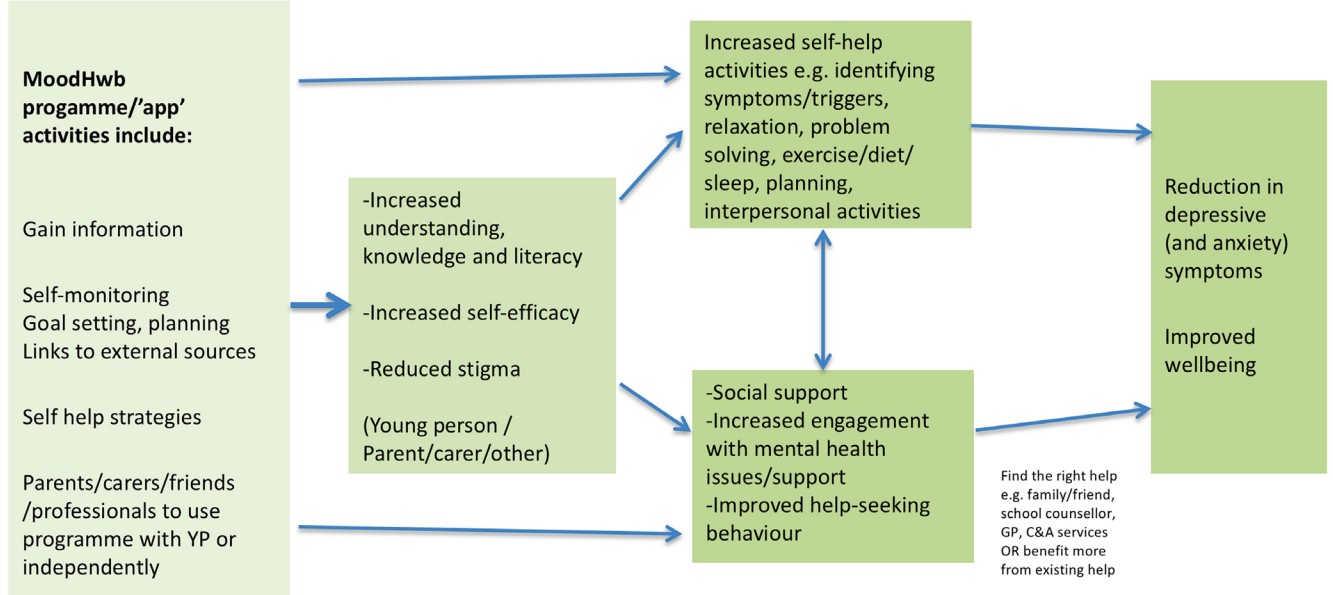

**Figure 2** Logic model for MoodHwb (above, from Bevan Jones et al[17] 2018b), including potential pathways (below).

programme acceptable to use and gave feedback on how it could be improved. Findings also provided initial support for the programme theory, particularly regarding depression literacy, self-efficacy and depressive symptoms. In line with guidance for complex interventions, a larger feasibility trial is now required.[21 22]

The primary aims of the current study are to further refine and evaluate the feasibility, acceptability and potential impact of the digital programme MoodHwb and the feasibility and acceptability of the trial methods.

## METHODS AND ANALYSIS
### Initial development phase
The first phase involves the further refinement of the digital programme, based on the feedback provided by participants during the earlier evaluation. The key

revisions recommended included improving the navigation, audiovisual and interactive elements, making it more personalised based on user preferences, developing the app version, and expanding the self-help and anxiety content.[20] There were subsequent consultations with practitioners at the proposed trial recruitment centres across Wales and Scotland, and researchers in the field.

Regular updates are required to the design and content of digital programmes because of the fast-moving pace of the field and the emerging evidence and guidelines. We have completed reviews of the codesign of digital technologies with children and YP[18] and technologies to support YP with depression and anxiety.[15] The principal investigator (PI) has visited research centres in youth and digital mental health in New Zealand, Australia and the Republic of Georgia. Workshops and meetings will be held with the digital media company and animator that helped to develop the original prototype.

As part of a pretrial acceptability phase, YP in Wales and Scotland (n=10–15) will be given access to the new prototype. They will be recruited through school counsellors, charities, youth advisory groups (YAGs) and self-referrals. Adolescents will be eligible to take part if they are aged 13–19 years and have sufficient understanding of the English language, access to the internet and a valid email address. They will be interviewed to ensure that the changes made are acceptable ahead

of the trial, and to explore issues of transferability in Scotland, given that the programme was developed in Wales. We will aim to recruit some YP who have experienced depressive symptoms. However, others with no experience of mental health difficulties will also give feedback, for example, on whether the programme is user-friendly and age-appropriate. Practitioners from health, education and youth services will be consulted further regarding the new prototype through meetings and online communication.

### Trial design
The feasibility study is a multicentre feasibility randomised controlled trial comparing the MoodHwb programme plus usual care (if the young person is receiving support) to digital information sheets on mood and well-being plus usual care (if receiving support) (figure 3). The research plan follows guidance for complex interventions,[21 22] and digital person-centred interventions.[23]

### Setting and participants
Trial participants will be identified by practitioners in schools (by eg, counsellors, nurses, teachers), mental health centres (mainly primary services), charities, youth services and sports academies in Wales, and in schools and charities via the second site at the University of Glasgow, Scotland. The School Health Research Network

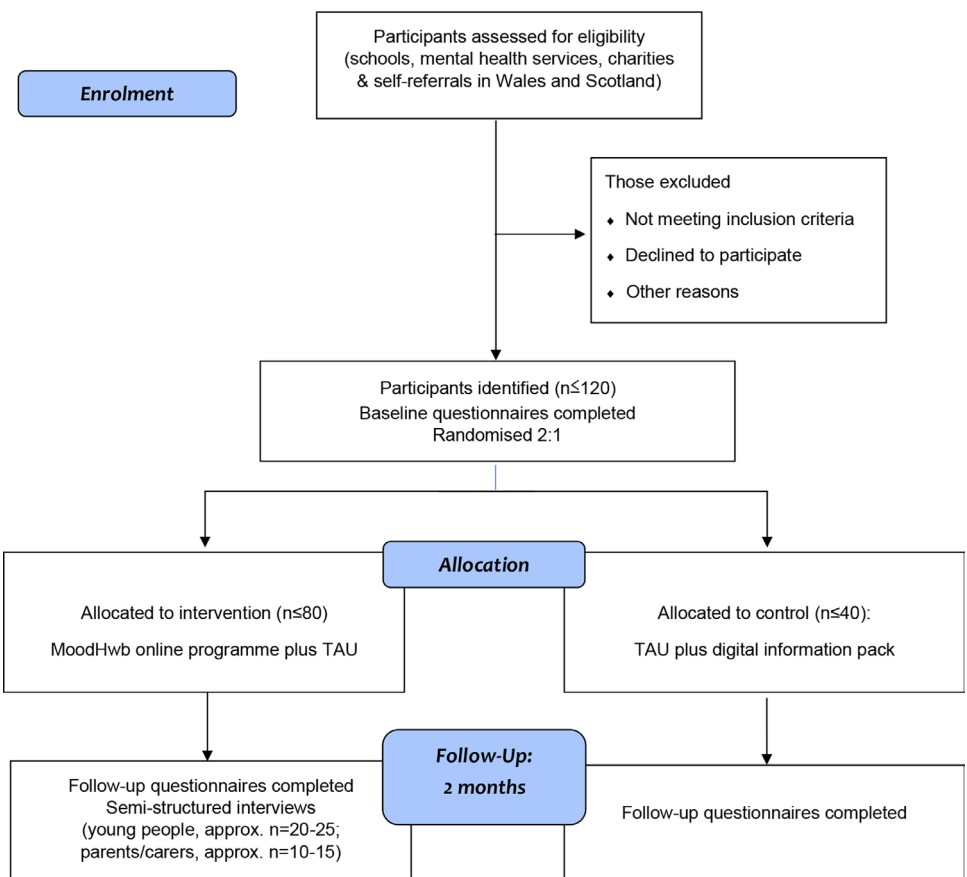

**Figure 3** Participant flow diagram. NB: Web/app usage data of those in the intervention arm will collected for 6 months after baseline.

(SHRN, Wales) and Schools Health and Well-being Improvement Research Network (SHINE, Scotland) will be involved. These sources were chosen as the developmental work suggested that the most appropriate setting for the programme would be the time when depressive difficulties start and YP first present—and it is therefore a form of early intervention.[17 20] We will also recruit self-referrals, for example, through social media and online advertising, including through the National Centre for Mental Health (NCMH), Wolfson Centre for YP's Mental Health (WCYPMH) and TRIUMPH youth mental health network.

YP will be eligible if they: (1) are aged 13–19 years, (2) are experiencing depressive symptoms, (3) have a sufficient understanding of English to complete outcomes measures, (4) have access to the internet and a valid email address and (5) live in Wales or Scotland. Exclusion criteria are: (1) already receiving specialist treatment for depression (eg, cognitive behavioural therapy (CBT) in secondary CAMHS) and (2) the presence of other mental health difficulties where alternative support might be appropriate. Parents and carers will be targeted for the trial if they have an adolescent with depressive symptoms. Professionals will be approached if they work with YP who present with mental health difficulties.

## Sample size

We aim to recruit up to 120 adolescent participants and their parents/carers. Approximately two-thirds of the participants will be from Wales and one-third from Scotland. We expect a dropout of around 30%, based on previous digital health studies and the development and early evaluation work. With a sample size of 120 we can estimate the overall expected retention rate of 0.7 to within ±0.082 (70%±8.2%).

## Recruitment

School, mental health, youth, or charity practitioners will identify potentially eligible YP and ask them whether they are interested in participating. The practitioners will provide a study postcard which directs the YP to the bilingual study website (www.ncmh.info/digital-support-study/), where there will be details of the study and information sheets. We will ask practitioners to keep a screening log of all YP approached. There will be a link to the website via social media and online advertisements for self-referrals.

If they are interested in taking part, YP will be asked to complete an online eligibility form on the website. This will include questions on their contact details and to assess whether they meet the eligibility criteria, including items from the Mood and Feelings Questionnaire (MFQ)[24] to measure depressive symptoms. They would be eligible if they score 20 or over on the MFQ. If they score 29 or over, they would also be advised to seek help and we will contact their general practitioner (GP), as this indicates a 'major depressive disorder'.[25] They will not be eligible if they access specialist treatment (eg, CBT; see exclusion criteria). The MFQ is commonly used in clinical and research practice and is a recommended screening tool in NICE guidelines for depression in children and YP.[8] The questionnaire and these cut-offs have also been used in previous trials of digital interventions for depression in YP.[26 27] The research team will then contact those who are interested to discuss the project (including consent forms and questionnaires), confirm their eligibility, respond to their questions and ask whether their parents/carers are interested in participating.

## Consent

Online consent forms are provided for the participants to complete, before completion of baseline questionnaires. If under 16 years old, YP will be asked to provide assent while their parent or legal guardian will be asked for consent. Dual consent and assent will be required for participation. Those aged 16 years or older will be able to provide their own consent. Parents/guardians and professionals who participate will also be asked to complete consent forms.

## Randomisation

Participants will be randomised at a 2:1 ratio to either the digital programme, MoodHwb plus usual care, or digital information sheets on mood and well-being in YP plus usual care (figure 3). We decided to randomise at a 2:1 ratio to allow for more information to be gained about those receiving the intervention. The permuted block randomisation will be computerised and stratified according to site (Wales and Scotland). This will be built into the study database overseen by the Centre for Trials Research (CTR, Cardiff University), providing remote access and preserving allocation concealment. Because of the nature of the intervention, participants will not be blind to their allocation, however, data analysis will be carried out blind to allocation.

## Intervention

MoodHwb is a digital programme to support YP with their mood and well-being, particularly those with elevated depressive symptoms. It can also be used by their families/carers, friends and practitioners. The programme and development process are described in detail in previous publications.[16–18]

It is designed to engage YP by using developmentally appropriate language, illustrations, animations and interactive components. It aims to promote self-help, help-seeking where appropriate and social support. While it was founded primarily on psychoeducation, it also includes elements of CBT, behavioural activation, positive psychology, and interpersonal, family systems and behaviour change theories. The first version was developed primarily as a website, although the interactive sections (eg, mood diary) are also available in an 'app'. It could be used independently as a 'standalone' programme by YP (or parents/carers, friends or professionals) or supported by another person.

From the welcome screen, the user is asked questions (eg, regarding their mood and anxiety levels), and the answers (1) are stored in their 'profile' section and (2) help to signpost to the relevant subsections on the subsequent 'dashboard' screen, therefore, personalising the content. Along with this 'mood diary', there are other interactive elements: a goal-setting element and a section to save links to helpful resources. Five sections were developed for YP (figure 1), and these cover: (1) low mood and depression, (2) possible reasons, (3) self-help approaches, (4) sources of help and (5) associated health difficulties. There is a sixth section to support those who might be concerned about YP, such as parents/carers, friends and practitioners. It is recommended that users navigate sections in this order, but they can use it in any way that is helpful for them. Each of these sections includes information, animations, personal stories, exercises and links to relevant resources. MoodHwb is bilingual—'hwb' is the Welsh translation for 'hub', and can mean a 'lift' or 'boost'. HwbHwyliau is the Welsh name for the programme.

Participants in the control arm will be emailed a digital information pack, which will consist of digital information sheets on mood and well-being (including depression) from the NCMH, Royal College of Psychiatrists and YoungMinds websites. Such resources offer general information, are widely accessible online, and have been used by control arm participants in similar trials.[27]

### Patient and public involvement
The programme was codesigned using a series of interviews (n=12) and focus groups (n=6) with (1) YP who either had a history of depression or were at elevated risk, (2) parents/carers and (3) professionals. Participants were recruited from CAMHS in Wales, and families from the EPAD study. As noted, further feedback on the programme was provided during the early evaluation.[20]

The project has been adopted by the Centre for Development, Evaluation, Complexity and Implementation in Public Health Improvement (DECIPHer, Cardiff University), and the study plans were developed in collaboration with their YAG, ALPHA (Advice Leading to Public Health Advancement). We will collaborate further with them as well as the YAGs of the WCYPMH, NCMH and TRIUMPH network. A young person has been invited to join the trial steering committee (TSC), and other YP (aged 13–16 years) have reviewed study documents, for example, information sheets and consent forms. Practitioners and researchers have been consulted regarding the plans.

### Assessment schedule
Data will be collected at (1) baseline and (2) follow-up, 2 months after randomisation. Data will be collected from (1) baseline and follow-up online questionnaires; (2) semistructured interviews with YP and parents/carers who used the programme; (3) a focus group with professionals and (4) Web/app usage data (table 1).

We will use remote methods for recruitment, consent and data collection (eg, phone, email, videoconferencing, online forms and questionnaires) because these have been shown to be efficient and effective ways of conducting research during the COVID-19 pandemic. This approach is consistent with the digital nature of the programme and would enable us to reach more YP. All participants will be offered £10 gift vouchers after (1) completing baseline questionnaires, (2) completing follow-up questionnaires and (3) participating in interviews.

### Baseline
At baseline, YP in both arms will complete questionnaires which will include items on their demographic and contact information, health and use of services, and standardised questionnaires on knowledge, stigma, self-efficacy, help-seeking, well-being, symptoms and life stresses (table 1). The parent/carer questionnaires will include items on the young person's details, use of services and symptoms, and their knowledge, beliefs, well-being, and symptoms. All participants will also complete items on the acceptability of the questionnaires.

### Follow-up
All baseline measures will be repeated at the follow-up time point, 2 months postrandomisation. In addition, those in the intervention arm will be asked about their use of the programme, and their views on the content and design. Those in the control arm will be asked whether they had access to MoodHwb, to explore contamination. A subgroup of YP (n=up to 20–25) and parents/carers (n=up to 10–15) who used the programme will be interviewed at follow-up on their use of and views of MoodHwb and the evaluation process. The Web/app usage of those using the programme will be monitored using Google Analytics up to 6 months postrandomisation, to provide information on longer-term use. There will be a focus group for professionals who work with YP from a range of settings, to ascertain their views.

### Process evaluation
The interviews with YP and parents/carers at follow-up and practitioner focus group noted earlier will provide more in-depth information on the acceptability and feasibility of the programme and evaluation process—in addition to the questionnaire and Web/app usage data. Participants will be purposively selected for interview based on their usage of MoodHwb (ie, frequent or occasional), age, gender, support received (eg, school, health service) and primary language (English, Welsh)—so that there is a broad range of viewpoints. A process evaluation framework has been developed, following MRC guidance,[28] and focuses primarily on the intervention but also examines certain trial processes. It addresses issues regarding reach, contamination, retention, recruitment, context, exposure and fidelity.

**Table 1** Schedule of enrolment, interventions and assessments for the trial

| Time point | Screening | Baseline (contact 1) | | Month 2 (contact 2) | |
| --- | --- | --- | --- | --- | --- |
| Activity | (contact 1) | Intervention | Control | Intervention | Control |
| Referral to the trial/register of interest | X | | | | |
| Eligibility assessment | X | | | | |
| Consent process | X | | | | |
| Randomisation | X | | | | |
| Access to MoodHwb | | X | | | |
| Questionnaires | | | | | |
| Adolescent Depression Knowledge Questionnaire[34] | | •/○ | •/○ | •/○ | •/○ |
| Depression Stigma Scale[35] | | •/○ | •/○ | •/○ | •/○ |
| General Self-Efficacy Scale[36] | | • | • | • | • |
| General Health-Seeking Questionnaire[37] | | • | • | • | • |
| Warwick-Edinburgh Mental Well-being Scale[31] | | •/○ | •/○ | •/○ | •/○ |
| Mood and Feelings Questionnaire[24] | | •/○ | •/○ | •/○ | •/○ |
| Revised Children's Anxiety and Depression Scale-Short Version-25[38] | | •/○ | •/○ | •/○ | •/○ |
| Hospital Anxiety and Depression Scale[39] | | ○ | ○ | ○ | ○ |
| Items on health economics and life events[40] | | •/○ | •/○ | •/○ | •/○ |
| Items on use and acceptability of MoodHwb | | | | •/○ | |
| Process evaluation | | | | | |
| Young person interview (n=20–25) | | | | • | |
| Parent/carer interview (n=10–15) | | | | ○ | |
| Professionals focus group | | | | * | |

(•Participant/○ parent/*professionals) 'contact' will comprise phone/videoconferencing discussion and completion of online forms/questionnaires.

## Outcome measures

### Primary outcome

The feasibility and acceptability of MoodHwb outcomes will include the level of usage and adherence (including number of times accessed and duration of use), views and acceptability of the design and content (including strengths, areas to develop further and approaches to integrate into services and charities), and technical and accessibility aspects. The feasibility and acceptability outcomes of the trial methods will include recruitment and retention rates, completeness of measures, representativeness, randomisation, contamination and acceptability (particularly of the remote methods). Progression criteria for a full trial have been developed with the TSC, based on these outcomes (online supplemental appendix 1).

### Secondary outcomes

Secondary outcomes assess the potential impact of the programme on a core set of domains, including depression literacy or knowledge, depression stigma, help-seeking behaviour, self-efficacy, well-being, and depression and anxiety symptoms. These will be measured via standardised validated questionnaires (table 1), and participant feedback (via questionnaires and interviews).

### Exploratory economic evaluation

We will investigate the feasibility and acceptability of collecting health economic data. We will identify and measure the resources involved in intervention delivery as well as resource use by participants, with reference to methodology from health economics guidance.[29 30] An estimation of the following will be collected: (1) intervention costs (development, hosting and maintenance costs), (2) health (eg, contact with GPs, hospitals and other health settings, medication), social, educational and personal resource costs, using questionnaire items adapted from the Client Service Receipt Inventory and (3) well-being, using the Warwick-Edinburgh Mental Well-being Scale.[31]

### Planned analysis

Primary outcome findings will be derived primarily from analysis of the questionnaires, interviews and focus group, with additional analysis of Web/app usage data regarding usage, adherence and technical aspects, and screening logs and participant database regarding recruitment and

retention. Secondary outcome findings will be derived primarily from analysis of validated questionnaires.

### Participant and web usage data

The flow of participants through the trial will be presented as a CONSORT (Consolidated Standards of Reporting Trials) flow chart.[32] Baseline demographic data will be presented for all consented and randomised participants. This will be summarised and tabulated for the whole group and by arm, using mean (SD) or median (IQR) where appropriate. All subsequent summaries will use complete cases and will employ the intention to treat principle (participants will remain in groups to which they were randomised irrespective of intervention received). Web usage data from Google Analytics will be tabulated, for example, related to number of times accessed, duration of use, sections accessed and language.

### Qualitative analysis

The interviews and focus group recordings will be transcribed, and data analysis will proceed using a thematic analysis approach. This is a process of identifying, analysing, reporting and interpreting patterns or themes, and is one of the most used qualitative approaches in health research.[33] Codes will be applied to broad themes and then broken down further into subcodes—30% will be double-coded to ensure the reliability of coding, and differences will be resolved through discussion. Transcripts will be closely examined to identify key themes and subthemes. Thematic analysis will be supported by the computer-assisted qualitative analysis software, NVivo.

### Quantitative analysis

Baseline questionnaire data will be scored according to the validated scoring algorithms from the relevant literature. Missing item-by-item data will be handled according to the scoring algorithm for each measure individually. Baseline scores will be presented descriptively, using summary statistics (means, SD, medians, IQR and percentages with associated 95% CI where appropriate) for the whole group and by arm. Distributional assumptions will be checked, and data transformation carried out if required. A comparison of randomised groups will be tabulated but not tested, in line with the feasibility aims. A comparison of baseline data for those completing and not completing the study will be tabulated to assess drop-out bias.

Scores at the 2-month time point will be tabulated and compared in a complete case analysis. No hypothesis tests will be carried out but 95% CI for the difference at 2 months between the two arms will be estimated and reported. These will be adjusted for baseline score and any factors such as site used to balance the randomisation. Statistical analysis will be performed using Stata software (version 17), but no p values will be presented. We will further develop and test the MoodHwb logic model (figure 2) using multiple methods, and examine possible mechanisms of action and active components. This will help to define a core set of outcome measures (related to the model) and identify the most appropriate primary outcome for evaluation in an effectiveness trial.

### Trial management

A trial management group (TMG) and an independent TSC have been established to monitor progress and advise the researchers. The TMG will meet approximately every 8–10 weeks during the trial planning and throughout the trial. The TMG is composed of the PI, coinvestigators, research assistant, trial administrator, CTR members and University of Glasgow site team. The TSC will meet approximately every 12 months. The TSC includes a young person, and an independent digital mental health researcher, statistician and mixed-methods researcher.

### Adverse event reporting and harms

For this trial, adverse events (AEs) are defined as any untoward medical occurrence in a participant administered an intervention and which are not necessarily caused by or related to that intervention. This would include significant deterioration in mental health symptoms, including depressive and anxiety symptoms, and emergence of thoughts of self-harm. Serious AEs (SAEs) include those resulting in death, are life-threatening, require hospitalisation or prolongation of existing hospitalisation, result in persistent or significant disability or incapacity, or other medically important conditions. The PI will assess each SAE for expectedness. A standard study/departmental procedure for documenting and dealing appropriately with these events will be followed. YP and parents/carers will be signposted to appropriate resources and services if there are concerns at the screening stage (see the Recruitment section) or from the questionnaire or interview responses if recruited. Their GP details will be collected at the screening and baseline stages, so they can be contacted. This is especially important, as the study will recruit self-referrals who might not access other support. An urgent safety measure is an action that the sponsor or PI may carry out to protect the participants of a trial against any immediate hazard to their health or safety—and must be notified to the REC immediately.

### Ethics and dissemination

Approvals were received from the Cardiff University School of Medicine REC and University of Glasgow College of Medical, Veterinary and Life Sciences REC for the pretrial acceptability phase with YP. Favourable ethical opinion for the overall study and trial was obtained from Wales REC 3 (reference 21/WA/0205), and approvals were received by HRA, HCRW, university health board R&D departments in Wales, and schools in Wales (including via SHRN) and Scotland (including via SHINE). The study was registered with ISRCTN (ISRCTN12437531). Approvals were received for MoodHwb to be added to the (Apple) App Store and Google Play App Store.

We will disseminate our findings to academics, researchers, policy-makers and practitioners through high-impact open-access publications and presentations at relevant academic, clinical and educational conferences and meetings. The findings will be disseminated online, for example, on the NCMH, WCYPMH and TRIUMPH websites. Results will be made available to all participants after the study completion.

## DISCUSSION

MoodHwb is a digital programme to support YP with their mood and well-being, especially those experiencing elevated depressive symptoms, and families/carers, friends and professionals who are involved in supporting them. It can be used as a website and as an app, as a standalone programme or supported by others, for example, in counselling or therapy sessions. Digital mental health technologies such as this have the potential to improve reach and access to therapies at a relatively low cost. MoodHwb will help to address the need for accessible, evidence-based technologies that are developed with user input and evaluated rigorously—and will provide YP and others with more options when looking for support.

### Strengths and limitations

MoodHwb was codesigned with YP with lived experience or at high risk of depression, parents/carers and practitioners. The programme is bilingual, interactive, personalised according to user needs and preferences, and multiplatform. It has been developed in line with evidence-based approaches, not only for YP but also for those concerned about them. Multiple methods will be used to collect data in a range of settings, including validated questionnaires, semistructured interviews, a focus group and Web/app usage monitoring. The involvement of participants in the second site in Scotland will help to boost recruitment and enable the exploration of transferability.

Given that this is a feasibility study, we will not be able to explore effectiveness; however, the information and experience will inform the design of a future effectiveness trial. While it is not feasible to blind participants to allocation; data analysis will be carried out blind to allocation. We have minimised the risk of contamination between trial arms by ensuring that a log-in is required to access MoodHwb. There will be items in the follow-up questionnaire for participants in the control arm on whether they have accessed MoodHwb during the study.

Having access to the internet and an email address is one of the inclusion criteria for the trial, however, some participants might not be adept at using digital devices (eg, mobiles, desktop computers), or might prefer other approaches (eg, face to face, printed). This might affect adherence to the digital programme and retention in the study, and we will explore methods of engaging participants during the study, for example, via questionnaires and interviews.

At the end of the study, there will be a period of reflection and planning regarding what further development, evaluation and implementation work are required for MoodHwb, both as a standalone and supported technology, in collaboration with YP. If it is ultimately found to be feasible and effective, it could be rolled out to help YP, families/carers and practitioners across a range of settings.

### Trial status

Recruitment started in November 2022 and will end in August 2023.

**Author affiliations**
[1]Division of Psychological Medicine and Clinical Neurosciences, Cardiff University, Cardiff, Wales
[2]Cwm Taf Morgannwg University Health Board, Rhondda Cynon Taf, Wales
[3]Wolfson Centre for Young People's Mental Health, Cardiff University, Cardiff, Wales
[4]Faculty of Medical and Health Sciences, The University of Auckland, Auckland, New Zealand
[5]Department for Health, University of Bath, Bath, England
[6]Centre for Trials Research, Cardiff University, Cardiff, Wales
[7]MRC/CSO Social and Public Health Sciences Unit, University of Glasgow, Glasgow, Scotland

**Acknowledgements** The authors would like to thank all the YP, parents, carers, professionals and researchers who participated in the development of this protocol. We would like to thank Caroline Warren, Gareth Watson, and Tim Pickles (all Cardiff University) for their contribution to the trial administration, database and statistics, respectively, and Emma McIntosh and Kathleen Boyd (both University of Glasgow) for their contributions to the health economics. We thank the TSC for their support.

**Contributors** RBJ designed and obtained funding for the study, with research support from SAS, FR, SM, PS and JIB, and advice from ER, RM, RP, SC, VP and RW-T (CTR). BW and AG provided research assistance at Cardiff University, and EH and MG coordinated the study at the University of Glasgow. RBJ drafted the initial study protocol. All authors contributed to and approved the final version.

**Funding** This work was supported by a National Institute for Health Research (NIHR) and Health and Care Research Wales (HCRW) Post Doctoral Fellowship programme, grant number NIHR-PDF-2018, awarded to RBJ. SAS was supported by UK MRC and Scottish Chief Scientist Office core funding as part of the MRC/CSO Social and Public Health Sciences Unit 'Complexity in Health Improvement' programme (MC_UU_12017/14 and SPHSU14) and MC-PC_13027. SM was supported by Cure Kids, New Zealand. The work was also supported by the NCMH, WCYPMH (established with support from the Wolfson Foundation) and TRIUMPH network (ES/S004351/1).

**Trial sponsor** Cardiff University is the Sponsor of the study.

**Competing interests** None declared.

**Patient and public involvement** Patients and/or the public were involved in the design, or conduct, or reporting, or dissemination plans of this research. Refer to the Methods section for further details.

**Patient consent for publication** Not applicable.

**Provenance and peer review** Not commissioned; externally peer reviewed.

**ORCID iDs**
Rhys Bevan Jones http://orcid.org/0000-0001-8976-9825
Sally Merry http://orcid.org/0000-0002-8281-1573
Paul Stallard http://orcid.org/0000-0001-8046-0784
Elizabeth Randell http://orcid.org/0000-0002-1606-3175
Bryony Weavers http://orcid.org/0000-0001-9654-3939
Anna Gray http://orcid.org/0000-0003-0546-6822
Rhys Williams-Thomas http://orcid.org/0000-0002-1779-3460
Vince Poile http://orcid.org/0000-0002-5770-5584
Rebecca Playle http://orcid.org/0000-0002-2989-1092
Jonathan I Bisson http://orcid.org/0000-0001-5170-1243
Rachel McNamara http://orcid.org/0000-0002-7280-1611
Frances Rice http://orcid.org/0000-0002-9484-1729
Sharon Anne Simpson http://orcid.org/0000-0002-6219-1768

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
