## [Reviewer comments · BMJ Open]

ARTICLE DETAILS

TITLE (PROVISIONAL)	Further development and feasibility randomised controlled trial of a digital programme for adolescent depression, MoodHwb: study protocol
AUTHORS	Bevan Jones, R; Merry, S; Stallard, Paul; Randell, Elizabeth; Weavers, Bryony; Gray, Anna; Hindle, Elaine; Gavigan, Marcela; Clarkstone, Samantha; Williams-Thomas, Rhys; Poile, Vince; Playle, Rebecca; Bisson, Jonathan; McNamara, Rachel; Rice, F; Simpson, Sharon

VERSION 1 – REVIEW

REVIEWER	Li, Sophie H Black Dog Institute
REVIEW RETURNED	01-Mar-2023

GENERAL COMMENTS	Thank you for the opportunity to review this manuscript describing the protocol for further development and a feasibility trial of MoodHwb. I found the paper provided sufficient detail to ensure transparency once the trial outcomes are reported. As I note the trial is underway, my comments are related to clarifying some aspects of the chosen methodology. 1. Could the authors provide justification for the selection of the control condition? From the manuscript, it appears that the content of the materials being delivered to the intervention and control conditions would be quite similar (i.e., predominantly psychoeducation).2. Will use of and engagement with the materials delivered to the control condition be measured and reported?3. Even though this is a feasibility trial, it may be helpful to clarify the primary objective of MoodHwb in supporting youth mood and wellbeing, e.g., is it to promote help seeking, depression awareness/literacy? There is some debate within the field regarding the benefit of increased MH awareness (https://doi.org/10.1016/j.newideapsych.2023.101010), so if this is one of the primary objectives of MoodHwb it may be worth commenting on the potential benefits of increased MH/depression awareness.
--

REVIEWER	Khan, Kareem University of Nottingham, Mental Health & Clinical Neurosciences
REVIEW RETURNED	08-Mar-2023

GENERAL COMMENTS	This protocol reports a feasibility study of a co-designed digital intervention called MoodHwb for adolescents with depression. The authors plan to refine the intervention further and then to conduct a multi-centre feasibility RCT comparing MoodHwb with usual care
--

	with a digital information pack plus usual care. The manuscript is well written, and the study is expertly thought out. My considerations for the authors are below. General comments 1. Why was the Mood and Feelings Questionnaire (MFQ) selected as the main outcome measure for depressive symptoms over others? Please could the authors clarify. Although this is not an effectiveness trial, they could also add what a clinically meaningful reduction in MFQ score would be. Background 2. Where it says, “Subthreshold depressive symptoms can also affect quality of life and are a risk indicator for depressive disorder”, do the authors have a reference for this, please? 3. A couple of the acronyms need spelling out (e.g. NICE and AACAP). 4. Whilst it is true that many young people in economically diverse countries have access to the internet and mobile devices, it is also true that some people are not adept at using technology as others. Thus, I feel a sentence or two explaining the ‘digital divide’ is warranted and how it may or not impact on your study. Methods and analysis 5. As this study will be recruiting patients via self-referrals, what safety net is in place for these patients whose mood deteriorates and perhaps even has an increase in suicidal thoughts and behaviours? A sentence or two on this is warranted. 6. Where the authors mention that one of the eligibility criteria is, “experiencing depressive symptoms”, how is this measured (e.g. via an outcome measure or clinical judgment)? Please clarify. 7. Dual consent and assent will be required for participation – however, what happens if the parent wants to participate, and the child does not? How will this be resolved?
--	--

VERSION 1 – AUTHOR RESPONSE

Reviewer: 1

Dr. Sophie H Li, Black Dog Institute

Comments to the Author:

Thank you for the opportunity to review this manuscript describing the protocol for further development and a feasibility trial of MoodHwb. I found the paper provided sufficient detail to ensure transparency once the trial outcomes are reported. As I note the trial is underway, my comments are related to clarifying some aspects of the chosen methodology.

1. Could the authors provide justification for the selection of the control condition? From the manuscript, it appears that the content of the materials being delivered to the intervention and control conditions would be quite similar (i.e., predominantly psychoeducation).

The information sheets/websites for participants in the control group offer general information on mood/wellbeing (not necessarily psychoeducation) and are already widely accessible online. Similar resources have also been used for other trials of digital interventions for depression in young people (e.g. Wright et al 2019 – see references section). We have added this information in the ‘Intervention’ section.

The MoodHwb content is more comprehensive, tailored for the user, and delivered via digital, multi-platform and interactive technology (via website and app). The original version was mainly a psychoeducational intervention with elements of cognitive behavioural therapy (CBT) and other

psychological and behavioural theories (Bevan Jones et al 2018 – see references section). The therapeutic (particularly CBT) content has been expanded further in the latest version of MoodHwb in response to the findings from the early evaluation (Bevan Jones et al 2020 – see references) and subsequent consultations.

2. Will use of and engagement with the materials delivered to the control condition be measured and reported?

We invite all participants to give feedback on their experiences of taking part in the project and using the programme/resources, but otherwise we do not specifically measure or report the use and engagement with the information sheets delivered to the control condition.

3. Even though this is a feasibility trial, it may be helpful to clarify the primary objective of MoodHwb in supporting youth mood and wellbeing, e.g., is it to promote help seeking, depression awareness/literacy? There is some debate within the field regarding the benefit of increased MH awareness

(<https://eur03.safelinks.protection.outlook.com/?url=https%3A%2F%2Fdoi.org%2F10.1016%2Fj.newidepsych.2023.101010&data=05%7C01%7Cbevanjonesr1%40cardiff.ac.uk%7C32e73dcff2ea4904c46a08db2169cff5%7Cbdb74b3095684856bdbf06759778fcbc%7C1%7C0%7C638140510567504448%7CUnknown%7CTWFPbGZsb3d8eyJWIjoiMC4wLjAwMDAiLCJQIjoiV2luMzliLCJBTiI6I1haWwiLCJXVCI6Mn0%3D%7C3000%7C%7C%7C&sdata=vOafCPZK3RKdHGILsU9fhD6%2B1y5AXM0f3HyW3iNllkY%3D&reserved=0>), so if this is one of the primary objectives of MoodHwb it may be worth commenting on the potential benefits of increased MH/depression awareness.

Thank you for the interesting reference. MoodHwb aims to promote self-help, help-seeking where appropriate and social support (see 'Intervention' section). The main outcome in an effectiveness trial would be determined by the findings from the feasibility trial, including the development and testing of the logic model (see 'Planned analysis' section). The primary outcome(s) would be selected mainly from the secondary outcomes of the current study i.e. knowledge/stigma, help-seeking, wellbeing and symptoms. We have not added further discussion on the potential benefits (or otherwise) of increased MH awareness as this might not be the primary outcome in a future trial – but we will keep the reference in mind for the future.

Reviewer: 2

Dr. Kareem Khan, University of Nottingham

Comments to the Author:

This protocol reports a feasibility study of a co-designed digital intervention called MoodHwb for adolescents with depression. The authors plan to refine the intervention further and then to conduct a multi-centre feasibility RCT comparing MoodHwb with usual care with a digital information pack plus usual care. The manuscript is well written, and the study is expertly thought out. My considerations for the authors are below.

General comments

1. Why was the Mood and Feelings Questionnaire (MFQ) selected as the main outcome measure for depressive symptoms over others? Please could the authors clarify. Although this is not an effectiveness trial, they could also add what a clinically meaningful reduction in MFQ score would be.

We selected the Mood and Feelings Questionnaire because this is commonly used in clinical and

research practice in child and adolescent mental health, it is a recommended screening tool in NICE guidelines for depression in children and young people (NICE 2019), and it has been used in other trials of digital interventions for depression in young people (e.g. Smith et al 2015, Wright et al 2019 – see references section). We have added a couple of lines on this in the 'Recruitment' section of the paper.

We also use the Revised Children's Anxiety and Depression Scale-Short Version (RCADS-25) in this trial, which includes items on depressive symptoms.

We will explore what a clinically meaningful reduction in score would be during the current trial, and use this information and findings from a literature review if the MFQ is used in a future effectiveness trial.

Background

2. Where it says, "Subthreshold depressive symptoms can also affect quality of life and are a risk indicator for depressive disorder", do the authors have a reference for this, please?

This is covered in part by Thapar et al's review on depression in young people for the Lancet (2022), cited in the first paragraph. We have also added the following reference: Bertha EA, Balázs J. Subthreshold depression in adolescence: a systematic review. *Eur Child Adolesc Psychiatry*. 2013 Oct;22(10):589-603. doi: 10.1007/s00787-013-0411-0.

3. A couple of the acronyms need spelling out (e.g. NICE and AACAP).

We have written these out in full in the 'Background' section.

4. Whilst it is true that many young people in economically diverse countries have access to the internet and mobile devices, it is also true that some people are not adept at using technology as others. Thus, I feel a sentence or two explaining the 'digital divide' is warranted and how it may or not impact on your study.

We have added a few lines on this in the 'Discussion' section.

Methods and analysis

5. As this study will be recruiting patients via self-referrals, what safety net is in place for these patients whose mood deteriorates and perhaps even has an increase in suicidal thoughts and behaviours? A sentence or two on this is warranted.

As noted in the 'Recruitment' section, we will contact the young person's GP if their MFQ score indicates a 'major depressive disorder'. We outline our approach to adverse events, such as a deterioration in mood, self-harm or suicidal thoughts/behaviours in the 'Adverse event reporting and harms' section. We have added a sentence regarding self-referrals in this section.

6. Where the authors mention that one of the eligibility criteria is, "experiencing depressive symptoms", how is this measured (e.g. via an outcome measure or clinical judgment)? Please clarify.

This is measured using the online eligibility form, which includes the MFQ - as noted in the 'Recruitment' section, and we have added a few words to clarify this in this section.

7. Dual consent and assent will be required for participation – however, what happens if the parent

wants to participate, and the child does not? How will this be resolved?

If a young person drops out of the study, the parent/guardian can continue to participate. However, a parent cannot take part from the start on their own, because the randomisation is based on the young person's participation.